# Insights from Murine Studies on the Site Specificity of Atherosclerosis

**DOI:** 10.3390/ijms25126375

**Published:** 2024-06-09

**Authors:** Godfrey S. Getz, Catherine A. Reardon

**Affiliations:** Department of Pathology, University of Chicago, Chicago, IL 60637, USA; reardon@uchicago.edu

**Keywords:** atherosclerosis, hemodynamics, site specificity, arterial tree, risk factors

## Abstract

Atherosclerosis is an inflammatory reaction that develops at specific regions within the artery wall and at specific sites of the arterial tree over a varying time frame in response to a variety of risk factors. The mechanisms that account for the interaction of systemic factors and atherosclerosis-susceptible regions of the arterial tree to mediate this site-specific development of atherosclerosis are not clear. The dynamics of blood flow has a major influence on where in the arterial tree atherosclerosis develops, priming the site for interactions with atherosclerotic risk factors and inducing cellular and molecular participants in atherogenesis. But how this accounts for lesion development at various locations along the vascular tree across differing time frames still requires additional study. Currently, murine models are favored for the experimental study of atherogenesis and provide the most insight into the mechanisms that may contribute to the development of atherosclerosis. Based largely on these studies, in this review, we discuss the role of hemodynamic shear stress, SR-B1, and other factors that may contribute to the site-specific development of atherosclerosis.

## 1. Introduction

While there are many excellent reviews on the initiation, progression, and regression of atherosclerosis, relatively few papers or reviews have been devoted to the development of lesions at particular sites of the vascular tree. In the decades that have followed our 2004 review on site-selective atherosclerosis [1], our understanding of the cellular and molecular aspects of atherogenesis has advanced immensely. This merits analysis of the improved appreciation of the basis for the focality of atherosclerotic lesion development.

Atherosclerosis is an inflammatory reaction that develops at specific regions within the artery wall of large and medium-sized arteries and at specific sites of the arterial tree over a varying time frame, with some reactions occurring early after the initiation of an atherogenic stimulus and some appearing later. There are a number of risk factors that are associated with the development of atherosclerosis, most of which are systemic in nature and therefore cannot on their own account for the focal nature of lesion development. Among the risk factors are hyperlipidemia, hypertension, diabetes, and sex. An ignored but critical systemic risk factor is age, which was highlighted in a recent analysis of clonal hematopoiesis [2]. These expanded blood cell clones with somatic mutations increase with age and produce enhanced levels of proinflammatory cytokines. One of the few good examples of the site-selective action of risk factors is seen in the multidisciplinary Pathological Determinants of Atherosclerosis in Youth (PDAY) studies. In these studies, early atherosclerosis was examined at autopsy in approximately 3000 apparently healthy (asymptomatic) individuals between the ages of 15 and 34 and its association with smoking, hypertension, hyperlipidemia, impaired glucose homeostasis (glycohemoglobin), and sex determined. Females had more abdominal aortic lesions than males, while the reverse sex bias was found in the right coronary artery [3]. Subjects who were smokers in life had three times as many raised abdominal lesions as nonsmokers, but smoking had no significant impact on lesions in the right coronary artery [4]. On the other hand, elevated glycohemoglobin levels were associated with more extensive lesions in the right coronary artery but not with the extent of abdominal aorta lesions [5]. The basis for these selective effects on atherosclerosis is not clear. Other examples of the effects of risk factors on site-specific lesion development are seen in the increased prevalence of peripheral vascular disease in patients with type III hyperlipoproteinemia (apoE2/E2) [6,7] and diabetes [8].

The focal nature of atherosclerotic lesions is generally thought to be driven by the hemodynamic properties of blood flow. Disturbed or oscillatory flow leads to low wall shear stress and occurs at sites where atherosclerotic lesions develop, while regions of the aorta with laminar flow with high wall shear stress seldom develop atherosclerosis. The geometry of the blood vessel contributes to the blood flow pattern. Disturbed flow is observed at the curvatures, branches, and bifurcation of arteries, and laminar blood flow is observed at straight, non-branching regions of the artery. The hemodynamic drivers of lesion development have largely been inferred from our understanding of the flow pattern in areas where lesions appear in human specimens. A good example of this is seen in the asymmetry of lesion development in human carotid arteries, where one artery contains significant atherosclerosis while the contralateral carotid artery is relatively plaque-free. Carotid arteries with lesions generally manifest lower shear stress [9]. But it is not clear whether the hemodynamic profile alone can account for lesion development at various locations along the vascular tree across differing time frames.

Since it is difficult to undertake detailed experimentation in humans, investigators have turned to experimental animals, large and small, to approach the mechanisms that account for the development of lesions at atherosclerosis-susceptible regions of the arterial tree [10,11]. Murine models are currently favored for the experimental study of atherogenesis. In a valuable commentary, Peter Libby points out that mice offer unique advantages for the exploration of the mechanisms of atherogenesis. These advantages relate to the relative ease of experimentally manipulating genes and controlling genetics, the environment, and other experimental variables, which cannot be applied in humans. While there are limitations to mice in relation to the complex, multifaceted pathogenesis of human atherosclerosis [12], nonetheless, studies in mice have contributed to the identification of cells, cellular pathways, and proteins that participate in atherogenesis, for example, inflammation and efferocytosis, for which targeted therapeutics have been developed [13,14].

Therefore, in this review, we focus primarily on studies in murine models. Since atherosclerosis develops in large and medium-sized arteries, we first outline the structure and components of these arteries and then describe a prototypic lesion as a platform for further analysis of lesion development. As the basis for insights into the mechanisms that contribute to site selectivity rests on temporal–spatial lesion development, we will describe the commonly used atherogenic mouse models of apolipoprotein E (apoE) deficiency and low-density lipoprotein (LDL) receptor deficiency and studies of lesion development over time. Finally, we end with a detailed treatment of what is known of the mechanisms that can be said to contribute to selective lesion development.

## 2. Artery Walls

The walls of large and medium-size arteries consist of three major layers, the endothelium, the media, and the adventitia, each of which participates in lesion development and may contribute to site-selective responses. The endothelial cells represent the thin barrier layer separating the blood from the rest of the artery. This layer is the most important sensor of the blood flow dynamics and hence is critically involved in the initiation of plaque development. Endothelial cells detect these shear stresses via mechanosensitive proteins such as G-protein-coupled receptors, flow-sensitive ion channels, and adhesion junction proteins, leading to the activation of cell signaling pathways that alter endothelial cell function and morphology in distinct ways [15]. The endothelial cells in disturbed (atherosusceptible) and laminar (atheroresistant) flow regions of the aorta differ in cell shape, the organization of the cytoskeleton, and intercellular junctional proteins. In the atheroresistant regions with laminar flow, the endothelial cells are aligned in the direction of the flow. Transcriptional and epigenetic differences have been noted in endothelial cells isolated from atherosusceptible and atheroresistant aortic areas and in cultured endothelial cells subjected to different flow patterns (see Section 5.1). For the most part, there is good concordance between the in vivo and in vitro results [16,17]. In vivo, the endothelium at atherosusceptible and atheroresistant sites sees the same systemic risk factors. 

Almost all the vascular smooth muscle cells of normal arteries are found in the media, where they contribute to the contractility and vascular tone of the vessel wall. Lineage mapping studies have shown that smooth muscle cells at various aortic sites along the arterial tree may originate from different embryological precursors [18,19]. The smooth muscle cells of the ascending aorta and the aortic arch and its branches are derived from neural crest cells, while the cells of the thoracic aorta are derived from somites, and cells in the abdominal aorta and those of its branches are derived from the serosal mesothelium. Not only do they arise from different sources but their postnatal signaling networks may also differ, which may contribute to differences in the response of the aorta sites to risk factors. For example, aortic homograft transplantation studies suggest that intrinsic differences in the aortic wall, including smooth muscle cells of different lineage sources, may contribute to the unique properties of the artery wall at different locations. In these studies, an atherosusceptible abdominal aortic segment was transplanted into the atheroresistant thoracic artery in canines, and vice versa (thoracic aortic section into the abdominal aorta) [18,20]. Following feeding them an atherogenic diet, the transplanted abdominal aorta segments developed severe atherosclerotic lesions despite being transplanted into an atheroresistant region of the aorta. Similarly, the transplanted thoracic artery segments developed minimal lesions when transplanted into the atherosusceptible abdominal aortic region. 

The adventitia is the outermost layer of the artery wall and has received increased attention [21]. The arterial adventitia is made up of many cell types, including fibroblasts, microvascular endothelial cells, resident macrophages, T cells, B cells, adipocytes, and perivascular nerves, and contributes to vascular development, homeostasis, repair, and disease pathologies. In humans and large animals, it is also the site that regulates the vasa vasorum, a microvascular network that supplies oxygen and nutrients to the cells in the outer layers of the vessel wall and the intima of developing plaques and can serve as a route for leukocyte trafficking into the lesion. During atherosclerosis, the adventitia accumulates inflammatory cells and is thought to be an important area of immune surveillance. The adventitia also contains resident vascular progenitor cells that in mice express the stem cell antigen 1 marker (Sca-1) [22] and that can differentiate into smooth muscle cells, endothelial cells, and macrophages. Adventitial progenitor cells have also been observed in humans. These cells may exchange between the adventitia and the media and contribute to lesion development and healing [23]. Whether these progenitor cells differ in concentration or function in the adventitia at different arterial sites is not known.

## 3. The Prototypic Atherosclerotic Lesion

The prototypic atherosclerotic lesion represents various phases of the chronic inflammatory reaction in arteries. As mentioned, the sites at which lesions develop are largely determined by the interaction of blood flow characteristics with the underlying vessel wall, especially endothelial cells. Dysfunctional and inflamed endothelial cells are found in regions of disturbed flow, especially in the presence of atherosclerosis risk factors. The most common systemic risk factor that drives the blood vessel response is hyperlipidemia, usually characterized by elevated LDL. The low shear stress of disturbed flow permits a longer residence time for LDL interaction with the arterial wall than in regions of lamellar flow. LDL particles, especially when its levels are elevated, cross the endothelium into the intimal space, where they are retained according to charged-based interactions with matrix proteoglycans like biglycan (human) or perlecan (mouse) [24,25,26] and are modified enzymatically and by oxidation and may aggregate [27]. The interaction of modified LDL with the endothelial cells enhances the expression of cell surface adhesion molecules and chemotactic factors to promote the influx of immune cells into the intima [28]. Monocyte chemoattractant protein 1 (MCP-1) is a key chemokine that promotes the ingress of monocytes. Intimal monocytes differentiate into macrophages and may proliferate locally under the influence of macrophage colony-stimulating factor (M-CSF). Macrophages also secrete MCP-1 to further enhance the ingress of monocytes. Within the intima, the resident and monocyte-derived macrophages take up modified LDL by the scavenger receptor pathways, phagocytosis, and micropinocytosis, resulting in the formation of lipid-loaded foam cells [27]. If the concentration of LDL is high enough, even unmodified LDL is taken up by micropinocytosis. The activation of macrophages in the intima results in the secretion of a panoply of cytokines and chemokines that act on local cells, promoting the further ingress of immune cells, including cells of the adaptive immune system, such as T cells, B cells, and natural killer T (NKT) cells [29,30]. These cells also contribute to the local cytokine milieu. The modified LDLs may serve as neoantigens, eliciting an immune response, with the production of antibodies against components of the atheromatous plaque [31]. It should be borne in mind that single-cell RNA sequencing (scRNA-seq) and cytometry by time of flight (CyTOF) analysis of murine atherosclerotic lesions have demonstrated the presence of multiple subtypes of macrophages in the atherosclerotic plaque, including some having characteristics of foam cells and others of a non-foamy inflammatory phenotype [32]. These studies also shed light on the heterogeneity of other leukocytes in lesions.

Foam cells may undergo apoptosis. In the initial stages of atherogenesis, these apoptotic cells are efficiently phagocytosed by neighboring phagocytes, mostly macrophages, according to a process designated as efferocytosis [33]. As lesions progress, efferocytosis becomes less efficient, and the accumulating apoptotic cells may undergo secondary necrosis, which, if extensive enough, produces the necrotic core of the advanced plaque. 

Growth factors secreted by cells in the developing lesion promote the migration of medial smooth muscle cells into the intima [34]. The intimal smooth muscle cells undergo transdifferentiation from a contractile to a synthetic phenotype and are a major source of matrix molecules, including collagen and the proteoglycans that play a role in retaining plasma lipoproteins in the intima [35,36]. The smooth muscle cells and collagen are major components of the fibrous cap that overlies the atherosclerotic plaque. An intact fibrous cap helps to stabilize the plaque, limiting its interaction with the blood coagulation system. However, the disruption of this cap by inflammatory macrophages may generate an unstable plaque, with the possibility of the formation of a thrombus, causing acute obstruction of the blood flow, with dire clinical consequences [34]. 

In addition, smooth muscle cells that migrate into the intima can contribute to the pool of lipid-loaded foam cells [37,38]. Indeed, lineage tracing studies have suggested that the proportion of smooth-muscle-derived foam cells may be equal to the proportion derived from macrophages [39]. However, smooth-muscle-cell-derived foam cells exhibit reduced phagocytic and efferocytic activity compared to macrophage-derived foam cells, which may contribute to the accumulation of apoptotic cells, with ensuing secondary necrosis in the lesions, indirectly contributing to necrotic core formation [40]. A recent scRNA-seq analysis of atherosclerotic lesions revealed the heterogeneity of the smooth muscle cells in the artery wall and also the expansion of various clones of smooth muscle cells [36,41]. This recalls the findings of Benditt that atherosclerotic atheromas are clonal expansions of smooth muscle cells [42].

The above is a broad overview of the typical process of atherogenesis, mediated by numerous molecules derived from local cell participants. However, the evolution of the atherogenic plaque is not a one-way pro-atherogenic process. It is balanced by counter-regulatory processes involved in lesion healing and resolution. This includes the removal of lipids from the plaque cells by reverse cholesterol transport [43], the emigration of macrophages from the lesions to draining lymph nodes [44], and the generation of resolving molecules, mostly from polyunsaturated fatty acids [45]. This may be promoted by the removal of pro-atherogenic stimuli such as hyperlipidemia and hypertension. The generation of alternatively activated macrophages, along with regulatory T cells, produces anti-inflammatory cytokines such as IL-10 and transforming growth factor-β (TGFβ), which can, in turn, stimulate the production of stabilizing matrix molecules [27,45,46].

The state of the atherosclerotic plaque represents a balance between pro- and anti-atherogenic processes, which may differ quantitatively from plaque to plaque and from site to site within the vascular tree. At any given time point, lesions at differing stages of evolution and resolution may be encountered. These considerations need to be borne in mind as we examine examples of apparent site-selective atherosclerosis in this review. 

## 4. Murine Models of Atherosclerosis

The above brief review of atherogenic lesion formation reveals an extremely complex set of interactions, with most of these potential interactions illuminated by studies in animals, particularly mice. Aside from mice, pigs, especially minipigs, are valuable models for atherosclerosis studies [11]. Inducing atherosclerosis in mice requires significantly increasing the levels of apoB-containing lipoproteins. Among the most frequently used models to study the mechanisms underlying atherosclerosis are mice with germ-line deletion of the apoE gene (*Apoe*) or the LDL receptor gene (*Ldlr*). 

The *Apoe*^−/−^ mouse was first described in 1992 by Maeda and colleagues [47] and by Plump and Breslow [48]. ApoE is an important ligand for the clearance of remnant lipoproteins. In its absence, remnant particles accumulate, leading to hypercholesterolemia and atherosclerosis on a low-fat chow diet. The administration of a high-fat, high-cholesterol, Western-type diet exacerbates hyperlipidemia and atherosclerosis. An important paper by Nakashima et al. [49] contributed to our thinking about the temporal–spatial evolution of atherosclerosis. To follow the evolution of lesions over time, atherosclerosis was examined throughout the vascular tree in *Apoe*^−/−^ mice maintained on chow or a Western-type diet for up to 40 weeks. Lesions were first noted in the aortic root, the lesser curvature of the aortic arch, the brachiocephalic artery (or innominate artery), and the right common carotid artery, followed by the carotid bifurcation, where lesions occurred on the outer wall of the bifurcation, while the flow dividers were spared [50]. Lesions were later observed in the branches of the superior mesenteric artery, the origins of the two renal arteries, and the aortic bifurcation and finally seen in the descending thoracic and common iliac arteries and the iliac bifurcation. The sites of lesion development were not different between the chow- and Western-type-diet-fed mice, but lesions developed more rapidly when the mice were fed the Western-type diet. In this study, sex did not influence the sites of lesion development. It is probably worth noting that these mice had a mixed C57BL/6 × 129 genetic background. 

The *Ldlr*^−/−^ mouse is a model of human familial hypercholesterolemia with elevated levels of LDL [51]. Humans with homozygous deficiency of the receptor develop xanthomatous lesions; widespread atherosclerosis, including in the aortic root; and often myocardial infarction and premature death [52,53]. In contrast to the *Apoe*^−/−^ mice, the development of significant atherosclerosis in the *Ldlr*^−/−^ mouse requires feeding them a high-fat diet [54,55]. This more readily allows us to study the initial stages of lesion development that occur after the initiation of a high-cholesterol diet. The two models also differ in their lipoprotein profile, with apoB100-containing LDL being the major lipoprotein in *Ldlr*^−/−^ mice and apoB48-containing remnant lipoproteins being the major lipoproteins that accumulate in *Apoe*^−/−^ mice. These two apoproteins differ in their binding affinity to intimal proteoglycans [56]. Due to these differences, the detailed early atherogenic mechanisms may well differ between the two models. Nonetheless, the distribution of atherosclerotic lesions is very similar for the two models as a function of age or duration of diet. 

These murine models have been used extensively to study the mechanisms underlying atherosclerosis. These studies use inbred strains of these murine models in specific pathogen-free environments fed a closely defined diet. While these experimental conditions allow for the performance of well-controlled experiments, they are not readily translated to free-living human environments. The advantage of mice is that multiple genes can be manipulated at the same time and their impact on atherosclerosis studied in a reasonable time frame. However, this has its limitations. The choice of the gene(s) for study may introduce an inherent bias. Often, the genes chosen are eliminated (knocked out) or overexpressed (transgenic or viral-mediated expression). These are extreme changes in gene expression that are not usually encountered in human populations. To ground these findings in the human experience, where more subtle quantitative or structural genetic variations are seen, Genome-Wide Association Studies (GWAS) are often consulted for corresponding human genetic variations. The closest murine counterpart is the Hybrid Mouse Diversity Panel (HMDP), which represents a panel of 100 inbred mouse strains studied on a hyperlipidemic background (transgenic for human apoE3 Leiden and human cholesteryl ester transfer protein) to identify natural variations in mice that contribute to atherosclerosis [57]. There is a good deal of concordance between the risk factors and trait loci identified in humans and mice. However, there is a lack of concordance for two aspects: namely sex and the effects of obesity [58]. In humans, atherosclerosis develops earlier in males, whereas in mice, females often exhibit greater atherosclerosis. Obesity in humans is associated with increased atherosclerosis, but atherosclerosis is reduced in genetically obese mice. 

However, not all studies examining the impact of specific genes or drugs on atherosclerosis in the *Apoe*^−/−^ and *Ldlr*^−/−^ models have produced the same phenotype [52]. This discordance may be related to mechanistic differences in the models, as mentioned above: the stage of atherogenesis at the time of sacrifice, the duration and nature of diet employed, the sex of the animals, or the housing conditions (e.g., microbiome complexity). 

## 5. Factors Contributing to the Site-Selective Development of Atherosclerosis 

In this section we will review what is known about the contribution of various factors listed in Figure 1 to the site specificity of atherosclerosis observed in murine atherosclerotic models.

### 5.1. Hemodynamics and Shear Stress

The progressive development of atherosclerotic lesions at different sites in the vascular tree might be a reflection of the extent of blood flow disruption, with it being greatest at the most rapidly developing sites. A strict quantitative comparison of flow dynamics with a propensity for lesion development through the vascular tree has not been reported, although a few attempts have been made. Giddens and colleagues [59] noted that the wall shear stress measured in the aortic arch and its principal branches is much higher in mice than in humans. The heart rate in mice is also much higher. The average flow velocity across the cardiac cycle (cm/s) in various aortic vessels in mice was measured, showing a progressive decline in flow velocity moving down from the suprarenal aorta to the infrarenal aorta and to the femoral artery [60]. This correlates with the relative temporal appearance of lesions in the mouse model, in which the suprarenal artery develops lesions sooner than the other sites. In another study, the flow pattern in the thoracic and abdominal aortas in Western-type-diet-fed *Ldlr*^−/−^ mice was changed from atheroresistant laminar flow to disturbed (oscillatory) flow using catheter-induced aortic regurgitation. The lesion burden in the descending thoracic and abdominal aortas were drastically increased, while the lesion burden in the non-flow-modified ascending aorta and the proximal aortic arch in the control and aortic regurgitation mice was comparable [61]. An increased prevalence of atherosclerosis in the descending thoracic aorta in humans with aortic regurgitation has also been observed [62]. These results highlight the importance of flow disturbance in lesion development. The genetics of the aortic geometry also highlights the importance of hemodynamics. A study of atherosclerosis in *Apoe*^−/−^ mouse strains with atheroresistant 129S6/SvEvTac (129) and atherosusceptible C57BL/6 genetic backgrounds found that the aortic root lesions were smaller in the 129 *Apoe*^−/−^ strain, as expected, but the aortic arch lesions were significantly larger compared to the C57BL/6 *Apoe*^−/−^ strain [63,64]. A possible contributor to the unexpected larger aortic arch lesions is that the aortic arch in the 129 strain has a sharper bend than in the C57BL/6 mice, which in modeling studies was associated with lower shear stress in the curvature of the aortic arch than in the C57BL/6 mice. Genetic crosses between these two strains show that the quantitative train locus (QTL) for aortic arch curvature and for atherosclerosis overlap [65]. 

Laminar shear stress upregulates mechanosensitive transcription factors (e.g., Krüppel-like factor (KLF)2, KLF4, nuclear factor-erythroid 2-related factor 2 (NRF2)) in endothelial cells. This leads to increased expression of endothelial nitric oxide synthase (eNOS) and bioavailable nitric oxide (NO), heme oxygenase 1, and thrombomodulin, among other genes, and KLF2-mediated inhibition of NF-kB to promote the anti-atherogenic, anti-inflammatory, and antithrombotic phenotype in the cells in laminar flow regions of the aorta [66,67]. The downregulation of KLF2 with the low shear stress of disturbed flow leads to the activation of NF-kB and the expression of inflammatory genes (e.g., vascular adhesion molecule (VCAM)-1, MCP-1, cytokines). Disturbed flow also leads to increased oxidative stress and endothelial–mesenchymal transition and induces metabolic reprogramming of the endothelial cells to increase glycolysis and reduce oxidative phosphorylation. Hypoxia-inducible factor 1α (HIF1α) appears to have an important role in this reprogramming. The induction of NAPDPH oxidase 4 (NOX4) and the subsequent increase in reactant oxidant species leads to the stabilization of HIF1α and subsequent induction of glycolytic enzymes such as hexokinase 2, (HK2), 6-phosphofructo-2-kinase/fructose-2,6-bisphosphatase 3 (PFKFB3), and 3-phosphoinosital-dependent kinase 1 (PDKI) (an inhibitor of pyruvate dehydrogenase) [68]. These metabolic changes also contribute to the increased inflammatory phenotype of endothelial cells subjected to disturbed flow. Additionally, the ATP-dependent purinergic receptor P2X7 may be implicated in the inflammatory phenotype of endothelial cells in regions with disturbed flow via increasing p38 signaling [69]. While this accounts for the location of atherosclerosis in particular regions of the aortic wall, whether differences in the expression levels of these proteins at different locations along the arterial tree account for site-specific atherosclerosis effects is not known. 

### 5.2. Genetics

The susceptibility to atherosclerosis of inbred strains of mice differs vastly. Cross-breeding strains of differing susceptibility to lesion formation clearly indicates the impact of genetics on atherogenesis. As mentioned above, the interaction of genetics and hemodynamics may explain the differential atherogenic response of the aortic root and the aortic arch in C57BL/6 *Apoe*^−/−^ mice and 129 *Apoe*^−/−^ mice [65]. Another genetic example is seen in relation to lesion formation in the innominate artery in *Apoe*^−/−^ mice with C57BL/6 and C3H/HeJ genetic backgrounds. The former strain is atherosusceptible, while the latter is atheroresistant, and this is reflected in the higher level of aortic root lesions in C57BL/6 *Apoe*^−/−^ mice. Yet innominate artery atherosclerosis is very similar for these two strains [70], indicating that the genetic impact on lesion development may be site-specific. CD44 was identified as a candidate gene associated with innominate artery atherosclerosis.

The literature is full of suggestions that the temporal–spatial atherosclerosis developmental profile is driven by the hemodynamic profile of the low shear stress of disturbed flow, which permits a longer residence time for effector particles within the arterial wall and produces differences in gene expression in the endothelial cells compared to laminar shear stress. While this is the consensual view, it does not seem to be the case in all studies [71]. For example, in a comparison of the evolution of lesions in *Ldlr*^−/−^ mice with C57BL/6 and FVB genetic backgrounds fed semi-synthetic diets with increasing levels of dietary cholesterol, ranging from 0% to 0.5%, there appeared to be more at play [72]. In the C57BL/6 *Ldlr*^−/−^ mice, their VLDL, IDL, and LDL levels were much higher than in the FVB mice with the diets with a higher cholesterol content. The C57BL/6 mice also had much larger aortic roots and innominate artery lesions than the FVB mice. On the other hand, the lesion sizes in the thoracic and abdominal aortas were similar for the two strains. Since various regions of the arterial tree are subject to the same level of lipoproteins, the differential responses of the abdominal aorta and the more proximal regions cannot be accounted for by the plasma lipids or hemodynamics. It is likely that there is a more complex explanation, probably dependent on the selective gene expression profile in the abdominal aorta.

### 5.3. LDL Homeostasis

Hyperlipidemia is a major risk factor for human disease and is required for lesions to develop in mice. Lipoproteins accumulate in the intimal of lesion-prone areas even before there is any evidence of the influx of blood cells. This was demonstrated morphologically using electron microscopy analysis by Frank and Breslow [73], who showed lipid particles the sizes of LDL, intermediate-density lipoprotein (IDL), or their remnants aligned along the collagen fibrils of the matrix of lesion-prone areas beginning at 3 weeks of age in chow-fed *Apoe*^−/−^ mice, while monocyte association with the endothelial cells was not observed until 5 weeks of age. Lipoprotein accumulation could be attributable to selective changes in lipoprotein permeability, influx, retention, or metabolism. Recent experiments have described an active transcellular transport pathway in endothelial cells that contributes to the accumulation of lipids in the artery wall. Caveolae and scavenger receptor class B, type 1 (SR-B1), which localizes to the caveolae, are more abundant in the endothelial cells in the atherosusceptible area of the aortic arch (lesser curvature) than in the atheroresistant area (greater curvature) [74]. LDL binding to SR-B1 promotes its interaction with dedicator of cytokine-like kinase 4 (DOCK4), leading to the activation of Rac1 and the transcytosis of LDL across the endothelial cells into the subendothelial intima [75]. The transcytosed LDL may be retained in the intima by binding to local proteoglycans [76]. Although most of the experiments by Huang et al. [75] were conducted with the aortic arch, SR-B1-dependent transcytosis is probably operative throughout the atherosusceptible areas of the aorta, as shown by the reduction in the en face staining of lesions in the whole aorta in *Apoe*^−/−^ mice with endothelial-cell-specific *Srb1* deficiency. The inhibition of SR-B1-mediated transcytosis of LDL may be a mechanism contributing to the atheroprotective effect of estrogen. The estrogen receptor GPER (G-protein-coupled estrogen receptor) reduced LDL transcytosis in cultured endothelial cells by downregulating the expression of SR-B1 [77]. 

Increased lipoprotein permeability through the endothelial layer may also contribute. Permeability has been probed in several ways. Aortic areas of increased permeability to macromolecules are visualized using an injection of albumin–Evans blue, a probe that can be used in experimental models in the absence of hyperlipidemia to provide a measure of in vivo sites of intrinsic increased permeability. Uptake of Evans blue dye in Yorkshire pigs was seen in the coronary ostia, the intercostal ostia, and the iliac bifurcation, sites of lesion formation [78]. Young and mature pigs exhibited similar patterns of staining. The exposure of vessels to the Evans blue complex ex vivo did not yield the same pattern of staining, highlighting the importance of hemodynamic factors. The blue areas were also the preferential sites of labeling with radioactive cholesterol. As mentioned, lipoproteins accumulated and were associated with the matrix in atherosusceptible murine sites early, before monocytes were observed to associate with the endothelial cells [73]. These findings could be accounted for by changes in the influx of lipoproteins, changes in their retention, or changes in their degradation or removal. This was partially examined in rabbit aortas without measurable atherosclerosis by Schwenke and Carew using radiolabeled LDL to measure the accumulation of intact LDL and tyramine-cellobiose-radiolabeled LDL to measure degraded LDL [79]. They found that the intact LDL concentration and degradation rates (per gram of fresh tissue) were several times higher in the aortic arch than in the descending thoracic aorta and abdominal aorta, which develop lesions later than the aortic arch. Examination of the LDL accumulation and degradation in the branched and non-branched regions of the abdominal aorta showed that the branched regions had a higher LDL concentration than the unbranched areas. However, the LDL concentration in the highly atherosusceptible aortic arch was significantly higher than in the branched regions of the abdominal aorta. What these results suggest is that the temporal–spatial development of lesions could be related, at least in part, to differences in the uptake and metabolism of LDL in atherosusceptible regions. The extent of disturbed flow in atherosusceptible arterial sites may be a determinant, but quantitation of this flow parameter in susceptible arteries is needed. Many other aspects of LDL homeostasis will need to be examined to fully understand its contribution to the temporal–spatial atherosclerotic developmental profile. Among these is the extent of LDL retention in the intima by proteoglycans and the extent of LDL modification by, for example, sphingomyelinase or oxidation.

### 5.4. LDL Oxidation

The oxidation of LDL is thought to be an important modification of LDL that facilitates the activation of macrophages in the early stages of atherogenesis. Treatment with antioxidants has hinted at the importance of this initial activation of macrophages. Surprisingly, treatment of *Apoe*^−/−^ mice with the antioxidant probucol led to an increase in the aortic root lesion area, though at other sites (i.e., the aortic arch, descending thoracic aorta, and abdominal aorta), there was a reduction in the lesion area [80]. The possibility that the oxidation of LDL is important at some arterial sites but not others has to be considered.

### 5.5. SR-B1

#### 5.5.1. SR-B1 and HDL

While SR-B1-dependent LDL transcytosis across the endothelium is an important factor that contributes to the difference between atherosusceptible and atheroresistant sites, the function of endothelial SR-B1 in the process of atherosclerosis is complex [81]. High-density lipoprotein (HDL) has several atheroprotective functions. Among them is that the interaction of HDL with SR-B1 in endothelial cells triggers binding of the adaptor protein PDZ domain-containing 1 (PDZK1) and the activation of AKT serine/threonine kinase 1 (AKT1), resulting in the activation of eNOS and increased production of NO, which is anti-atherogenic [82]. In addition, one of the atheroprotective mechanisms of HDL might be to compete with LDL for SR-B1-dependent LDL transcytosis [75]. 

SR-B1-mediated transcytosis of HDL by the endothelial cells [75] may also enhance reverse cholesterol transport, in which the interaction of intimal HDL with ATP-binding cassette G1 (ABCG1) in macrophage foam cells promotes the efflux of cholesterol to HDL for its transport to the liver [83]. SR-B1 on hepatocytes mediates the selective uptake of cholesteryl esters from HDL and influences HDL composition [81]. 

#### 5.5.2. SR-B1 and Coronary Artery Atherosclerosis

As valuable as the “mighty” mouse is for investigating the mechanisms involved in atherogenesis, this model does have important limitations in extrapolating its results to humans [12]. A major limitation is the absence of significant coronary artery lesions in most studies involving *Apoe*^−/−^ and *Ldlr*^−/−^ mice. Yet certain complex genetic models can elicit occlusive coronary disease accompanied by myocardial infarction. For example, in Western-type-diet-fed mice with combined *Apoe* and *Ldlr* deficiency [84] and with overexpression of the urokinase plasminogen activator in the macrophages in *Apoe*^−/−^ mice fed a Western-type diet [85], the development of occlusive coronary arterial disease and myocardial infarction occurred. The most dramatic murine model for occlusive coronary artery atherosclerosis with myocardial infarction and notably premature mortality (mean age of 6 weeks) is seen with chow-fed mice with double knockout of the *Apoe* and *Srb1* genes [86]. *Ldlr*^−/−^*Srb1*^−/−^ mice fed a high cholesterol diet also develop coronary artery atherosclerosis and myocardial fibrosis [87]. Mortality is particularly premature when the diet is supplemented with sodium cholate. Interestingly, in contrast to what appears to be the response of the aortic root to probucol treatment [80], coronary artery atherosclerosis in *Apoe*^−/−^*Srb1*^−/−^ mice is rescued by probucol treatment [88]. Obstructive coronary artery disease being observed in mice with global deficiency of *Srb1* together with *Apoe* or *Ldlr* deficiency suggests that in the coronary arteries processes other than LDL transcytosis and eNOS activation may be at play [86,87]. Paracellular diffusion of LDL from the plasma may be one such process. 

Individual global knockout of the pathway by which HDL binding to SR-B1 promotes NO production (*Pdzk1*, *Akt1*, or *eNOS*), accompanied by *Apoe* deficiency, also promotes occlusive coronary artery disease but only when the animals are fed atherogenic diets, with some requiring cholate in the diet [89,90,91]. Bone marrow transfer experiments indicate that it is the deficiency of *Akt1* in non-hemopoietic cells, probably the endothelial cells, that accounts for the coronary artery phenotype in *Akt1*^−/−^*Apoe*^−/−^ mice [90].

A related but valuable model for the study of coronary artery disease and myocardial infarction is *Srb1*^−/−^ mice crossed with a hypomorphic apoE mutant (threonine 61 substituted with arginine), in which apoE is expressed at 5% of the wild-type levels and fed a high-fat, high-cholesterol, cholate-containing diet [92]. SR-B1 is expressed on endothelial cells, macrophages, and hepatocytes. The importance of SR-B1 outside of the endothelium is indicated by the observation that the transplantation of bone marrow from apoE hypomorphic mice expressing SR-B1 into *Srb1*^−/−^ apoE hypomorphic mice protects against coronary artery disease [93]. The anti-atherogenic role of SR-B1 in the macrophages may be related to its role in promoting cholesterol efflux, reducing their inflammatory phenotype and maintaining efferocytotic activity [81]. 

But the transplantation of bone marrow deficient in *Srb1* into *Ldlr*^−/−^ recipients suggests that the role of SR-B1 in bone-marrow-derived cells is complex. SR-B1-expressing bone marrow reduces very early atherosclerosis after 4 weeks of a Western-type diet, yet at longer periods on this diet, atherosclerosis is increased [94]. This suggests that SR-B1 on bone-marrow-derived cells, likely macrophages, can be pro-atherogenic or anti-atherogenic at different stages of atherogenesis. The expression of SR-B1 on macrophages is associated with reduced expression of the inflammatory cytokines IL-1, IL-6, and tumor necrosis factor-α (TNFα) [95]. 

In summary, SR-B1 is associated with pro- and anti-atherogenic pathways depending on context and the site of expression.

### 5.6. Macrophages

Intimal lipoproteins promoting atherosclerosis requires the participation of the macrophages. In the absence of M-CSF, which regulates the proliferation and survival of monocytes/macrophages, very little atherosclerosis is seen in *Apoe*^−/−^ [96] and *Ldlr*^−/−^ mice [97]. In the highly atherosusceptible, lesser curvature of the ascending aortic arch, intimal leukocytes, with markers of monocytes or dendritic cells, are present in normal arteries. In the first few days after the initiation of feeding a cholesterol-rich diet to *Ldlr*^−/−^ mice, these cells become loaded with lipids [98]. These foam cells are not clearly evidenced in less atherosusceptible regions like the greater curvature of the ascending aortic arch, the descending thoracic aorta, or the abdominal aorta, even though the latter is a site of atherosclerosis after exposure to a cholesterol-rich diet for longer durations [99]. These cells have the characteristics of dendritic cells [100]. After the consumption of a cholesterol-rich diet for 2–12 weeks, foam cells are derived from monocytes recruited from the blood and from the local proliferation of macrophages [99]. 

The influx of monocytes into evolving lesions requires chemokines and their receptors. In the combined absence of CCL2, CCR5, and CX3CR1, very few lesions develop [101]. The absence of CX3CR1 in the *Apoe*-deficient background results in reduced atherosclerosis in the aortic root and aortic arch and reduced en face staining of lesions in the whole aorta [102,103]. In mice, there are two major monocyte subtypes, Ly6C^hi^ and Ly6C^lo^. Ly6C^hi^ monocytes are more inflammatory and preferentially infiltrate the artery wall [104]. The monocyte subsets differentially employ the three receptors to enter the artery wall [105]. Ly6C^hi^ monocytes require all three receptors, while the influx of Ly6C^lo^ monocytes is much less dependent on CX3CR1 and partially dependent on CCR5. These chemokine/chemokine receptor pairs are not all equally involved in atherogenesis at all arterial sites. For example, in the absence of the ligand CX3CL1, lesions are preferentially reduced in the innominate artery in *Apoe*^−/−^ mice [106].

### 5.7. Adaptive Immune System

The adaptive immune system embodies several subsets of B and T cells. We examined chow-fed male *Apoe*^−/−^ mice lacking both B and T cells due to deficient of recombination activating gene 1 (*Rag*^−/−^ mice) at 27 weeks of age. When adjusted for plasma cholesterol, aortic root lesions were reduced in the immune-deficient mice, though no differences were observed in the innominate artery [107]. This might be explained by the different balance between pro-atherogenic and anti-atherogenic immune cells (e.g., CD4+ Th cells vs. CD4+ regulatory T cells) at the two sites, at least at the time of monitoring. This suggestion has yet to be validated. These animals were fully backcrossed into the C57BL/6 background. In this background, Western-type-diet-fed *Ldlr*^−/−^ mice showed similar results. However, in a strain that was only 93% C57BL/6, adaptive immune deficiency was associated with a reduction in lesions in the aortic root, as well as in the innominate artery [108]. This suggests that the genes expressed in the innominate artery may influence the state of regulatory T cells at this site.

Other cells of the immune system also have a site-specific impact on atherosclerosis. NKT cells exist as at least two subtypes [109]. Type I or invariant NKT cells express a semi-invariant T cell receptor (TCR) composed of the Vα14-Jα18 chain coupled with a limited number of TCRβ chains in mice. Type II NKT cells have more variable TCRs. Both recognize the lipid antigens presented by CD1d on antigen-presenting cells. *Ldlr*^−/−^ mice transgenic for the Vα14-Jα18 TCRα chain with increased levels of type I NKT cells and mice lacking type I NKT cells due to deficiency of the Jα18 chain were fed a Western-type diet for 4 or 12 weeks [110]. The atherosclerosis response was quite complex. There was a selective decrease in atherosclerosis in the ascending aorta near the lesser curvature of the aortic arch in the absence of type I NKT cells after 4 weeks of the diet (early atherosclerosis) despite a noticeable increase in plasma lipids. But after 12 weeks of the diet, the lesions were similar at this site. The overexpression of Vα14 Jα18 resulted in a site-specific increase in innominate artery atherosclerosis after 12 weeks of feeding. 

Lymphotoxin ligands are mainly expressed by lymphocytes and innate lymphoid cells and come in two forms [111]. Soluble homotrimers of lymphotoxin α ligands (LTα_3_) are secreted mainly by lymphocytes and signal via TNF receptors, which are widely expressed. The heterotrimeric LTα_1_β_2_ ligand is bound to the cell surface of lymphocytes and signals through interaction with the lymphotoxin β receptor (LTβR), expressed on the cell surface of many cell types. Female *Ldlr*^−/−^ *Lta*^−/−^ animals that lacked both lymphotoxin ligands were compared with mice with T-cell-specific lymphotoxin β deficiency, lacking LTα_1_β_2_ on their T cells, and mice deficient in the LTβR. The animals were fed a Western-type diet for either 6 or 12 weeks. Atherosclerosis in the innominate artery was not impacted by the manipulation of this ligand–receptor pathway at either time point. There was a selective reduction in aortic arch lesions with global deficiency of *Lta* or T-cell-specific deficiency of *Ltb* at the 12 week time point. The aortic root lesion responses were quite complex. Aortic root lesions were reduced in *Lta* deficiency, associated mainly with reduced atherosclerosis in the left-coronary-artery-associated sinus, which is the sinus that carries the major burden of lesion development compared to the right-coronary-artery-associated sinus and the sinus not associated with a coronary artery. However, in T-cell-specific *Ltb* deficiency, the aortic root lesions were increased, especially in the non-coronary-artery-associated sinus. In contrast, compared to the *Ldlr*^−/−^ mice, the *Ltbr*^−/−^*Ldlr*^−/−^ mice exhibited no lesion differences at any arterial site examined. In the *Apoe*^−/−^ background with Western-type diet feeding, the absence of the LTβR was associated with a reduction in lesions in the aortic root and reduced en face staining of the whole aorta [112]. Bone marrow transplantation studies suggest that the expression of the receptor on hematopoietic cells, likely monocytes, is determinative. Clearly, this is a very complex system, and much further experimentation is required to unravel the intricacies of the observed findings.

In aged chow-fed *Apoe*^−/−^ mice, a tertiary lymphoid organ develops in the adventitia surrounding abdominal aortas with atherosclerotic lesions [113]. The lymphoid organs are rich in B cells and T cells, including regulatory T cells. The tertiary lymphoid organs begin to appear at 32 weeks of age, being obvious at 52 weeks and peaking at 78 weeks. Their development is dependent on signaling via the LTβR on the smooth muscle cells, resulting in the secretion of the lymphorganogenic chemokines CXCL13 and CCL21 [114]. Vascular smooth muscle cell depletion of *Ltbr* suggests that the signaling via the receptor is age-dependent, initially having no impact on atherosclerosis in young mice and then being atheroprotective in aged mice, especially in the abdominal aorta [115]. Intriguing is the territoriality of these tertiary lymphoid organs, which are particularly found around the abdominal aorta, a site that develops lesions later in the temporal progression of atherosclerosis in the vascular tree. The basis for this specific location is yet to be clarified.

### 5.8. Toll-Like Receptors (TLRs) and Lesion Localization

Many TLRs have been implicated in atherogenesis. In relation to differential lesion localization, TLR2 is particularly relevant. TLR2 is widely expressed among the cells involved in atherogenesis, including endothelial cells, monocytes and macrophages, smooth muscle cells, and lymphocytes [116]. TLR2 is upregulated in the endothelium by disturbed flow and hyperlipidemia [117]. TLR2 can be activated by both endogenous and exogenous activators of bacterial origin [118]. Among the endogenous activators are oxidized LDL, apoCIII, biglycan, versican, decorin, and hyaluronan fragments. The synthetic TLR2 activator PAM3CSK4, in a dose- and TLR2-dependent fashion, was shown to enhance atherosclerosis in an *Ldlr*^−/−^ model fed a Western-type diet, particularly in the abdominal aorta [119]. The basis for the almost unique impact at this site is unclear. However, it is in part dependent on the expression of TLR2 in bone-marrow-derived cells. Among the effects of TLR2 activation is the proliferation of regulatory T cells, which is accompanied by temporary inhibition of their suppressive activity [120].

### 5.9. Perivascular Adipose Tissue (PVAT)

Recent work has indicated the potential importance of PVAT in the evolution of atherosclerosis. PVAT is located adjacent to the adventitia of most of the major arteries, with the cerebral and pulmonary arteries being notable exceptions, and is functionally distinct from other adipose depots [121]. It is composed of adipocytes, nerves, monocytes, endothelial cells, macrophages, and T and B cells. PVAT regulates vasomotor tone and vascular homeostasis through the secretion of relaxing and contracting factors and inflammatory status through the secretion of pro- and anti-inflammatory factors. The regional phenotypic heterogeneity of perivascular adipocytes has been noted among different PVAT arterial sites. In rodents, thoracic aorta PVAT is composed of mostly brown adipocytes, abdominal aorta PVAT contains predominately white adipocytes with some brown adipocytes, and mesenteric PVAT is largely white adipocytes [121,122]. In humans, the perivascular fat contains mostly white adipocytes. Interestingly, TLR2 is upregulated in brown adipose non-PVAT tissue in obese mice [123]. 

The potential contribution of PVAT may differ depending on the vascular bed it surrounds. A recent study has explored this complexity in the thoracic aorta in mice. The anterior thoracic and lateral thoracic artery PVAT pads differ in their developmental pathways and transcriptional profiles and are associated with different effects on vascular function [124].

## 6. Conclusions and Future Directions

From this review, it is clear that atherosclerosis and its site specificity is extremely complex and that the basis for the temporal–spatial lesion localization in *Apoe-* and *Ldlr*-deficient mice is incompletely understood. While most studies of regionalization have been made in mice, there is also evidence that this occurs in humans too, as evidenced by the PDAY studies [1]. In a great deal of the research on the mechanisms of atherosclerosis using mouse models, the emphasis is on the lesions in the aortic root, a site not often developing lesions in humans, and on en face analysis of lesions over the entire aorta and principal branches, without specific analysis of discrete aortic regions. To achieve a better understanding of the mechanisms of these localizations, further studies are necessary at appropriately selected sites at various stages of atherogenesis. These include a more accurate measurement of the flow patterns along the extent of the aorta and its branches, a detailed analysis of transcytosis, and the distribution of proteoglycans along the vasculature to determine the basis for lipoprotein retention at these sites. In addition, the application of evolving single-cell technologies, such as scRNA-seq, CITE-seq (cellular indexing of transcriptomes and epitopes), scATAC-seq (single-cell transposase-accessible chromatin with sequencing), and CyTOF, to analysis of atherosclerosis at different sites in the vascular tree should provide insight into the cellular and molecular heterogeneity that may contribute to the site-specific development of atherosclerosis [125]. This is a tall order, but this will be required to understand the patterns of lesion initiation and progression along the vasculature.

## Figures and Tables

**Figure 1 ijms-25-06375-f001:**
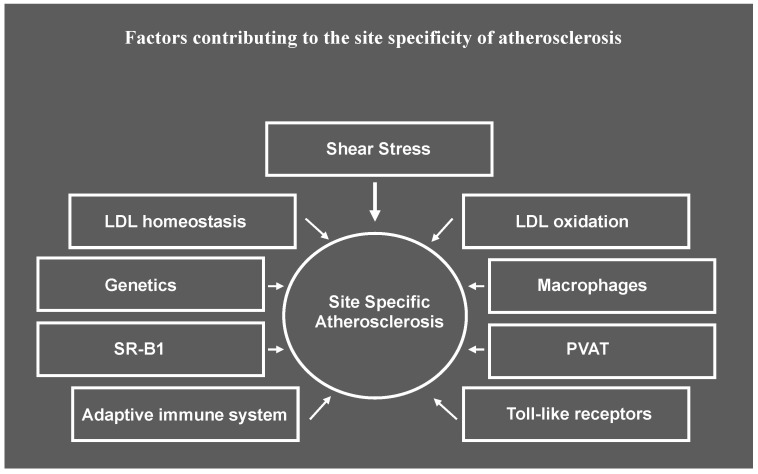
Factors contributing to site specificity of atherosclerosis. While shear stress induced by blood flow has an important role in priming cells in the artery wall for susceptibility or resistance to atherosclerosis, the response to risk factors at atherosusceptible sites may be differentially impacted by other factors that result in lesion initiation and progression proceeding at different rates at different sites along the artery wall. SR-B1, scavenger receptor B1, PVAT, perivascular adipose tissue.

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
