# Peer review of "Insights from Murine Studies on the Site Specificity of Atherosclerosis"

_ijms, 2024, doi:10.3390/ijms25126375_

Round 1

Reviewer 1 Report (New Reviewer)

Comments and Suggestions for Authors

This is an elegantly written and interesting manuscript by Getz and Reardon, who are experts in writing this type of narrative review. The topic is timely and highly relevant. In recent years, there have been breakthroughs in our understanding of the site-specificity of atherosclerosis.

I have a few comments that the authors might want to consider:

In lines 240-241, "the development of atherosclerosis in the Ldlr-/- mouse requires the feeding of a high-fat diet." See the development of atherosclerosis on a chow diet in Ldlr-/-, which I find significant, in the references:

- Ishibashi S, Goldstein JL, Brown MS, Herz J, Burns DK. Massive xanthomatosis and atherosclerosis in cholesterol-fed low-density lipoprotein receptor-negative mice. J Clin Invest. 1994; 93:1885–1893. doi: 10.1172/JCI117179

- Du W, Wong C, Song Y, Shen H, Mori D, Rotllan N, Price N, Dobrian AD, Meng H, Kleinstein SH, et al. Age-associated vascular inflammation promotes monocytosis during atherogenesis. Aging Cell. 2016; 15:766–777. doi: 10.1111/acel.12488

In line 268, it would be helpful if the authors could explain the lack of concordance between atherosclerosis and the effects of obesity in mice.

Measuring lesions in various parts of the arterial tree is possible in mouse models and could be important for discussing site specificity. However, it is important with some methodological considerations before concluding that there are "no lesion differences." The study should be designed and powered to draw these conclusions. Small differences could be missed, or false positive outcomes might be reported. The recommendation in lines 274-277, "Given this complexity, it is necessary with any selected intervention to measure lesions at multiple sites, at various stages of lesion development, and in animals of both sexes, as well as attempts at duplication in several vivariums", may not be universally feasible. Designing experimental atherosclerosis studies to allow for adjustments for multiple statistical testing might not be recommended if the study addresses a specific scientific question, given animal ethical guidelines. The authors provide a more balanced recommendation at the end of the manuscript in the "Conclusions and Future Directions" section.

In line 304, "the descending thoracic artery" should probably be corrected to "the descending thoracic aorta."

Author Response

Reviewer 2 Report (New Reviewer)

Comments and Suggestions for Authors

Getz and Reardon explore the existing knowledge about mechanisms of atherosclerosis obtained from tow major murine models, which have been employed for several decades.

In the beginning of the review, the authors praise the mouse models as valid for understanding the mechanisms of human atherosclerosis. In the conclusions however, they point out multiple crucial differences between mice and humans, including cardiovascular hemodynamics. So I think it will be interesting to hear from the authors what is there opinion on overall advancements in understanding mechanism of human atherosclerosis by using mouse models. Is there any existing or conceptual therapy in sight to reverse atherosclerotic lesion of the vessel wall derived from the decades of experimental work? This has to be pointed out more prominently.

What about ApoE/LDLr double-knockout mice? Do they show similar mechanistic and phenotypic atherosclerosis as the single knockouts? 

Author Response

This manuscript is a resubmission of an earlier submission. The following is a list of the peer review reports and author responses from that submission.

Round 1

Reviewer 1 Report

Comments and Suggestions for Authors

The review is a nice overview about common concepts that may explain the sitie-specifitity of atherosclerosis.

Major comments:

In the abstract and introduction the authors promise things that are not clearly addressed in the manuscript. First, the abstract does not give any specific pathway that may explain site-specificity. Instead of this, the authors promise to discuss what is known. That is however not the case. They do not discuss common concepts they describe these concepts. The review would significantly be improved if the authors can come up with a final concept that they believe summarizes our current understanding of the topic at least in the summary. The authors clearly state that they consider this review as an up-date of their 2004 review. However, it is never mentioned what the differences in the concepts of 2004 and 2024 is. At which point did new experimental studies change the view from 2004? This would improve the review significantly. Are there any new conclusions that potentially will affect clinical concepts in the future? The authors state that there is a difference between small and large animals and properly a significant impact on the translation to human and clinical relevant questions. However, there is only a chapter on mice and in the conclusion the authors state that mice studies were significant for humans. These conflicting statements lead to confusion. Please clarify where the small animal models have an advantage and where the large animal models are required.

Fig. 1 is not informal. The box for hemodynamic may account for either shear stress or blood pressure. Please specify. A legend is missing for the Figure. I.e., please define PVAT etc.

What is a PDAY study? Abbreviation neither explained in the list of abbreviations nor is the abbreviation part of the title of ref. 1.

Are you talking about sex or gender in your review?

Please replace the unit “cm/sec by the SI unit “m/s”.

Reviewer 2 Report

Comments and Suggestions for Authors

See the attachment.

Comments on the Quality of English Language

Minor editing of English language required

Round 2

Reviewer 2 Report

Comments and Suggestions for Authors

The work is still poorly organized and doesn't reflect the aim of the review, namely a work aims to explore the factors that may contribute to this site-specific development of atherosclerosis.

Comments on the Quality of English Language

Minor editing of English language required